# Chemical Composition of *Pinus nigra* Arn. Unripe Seeds from Bulgaria

**DOI:** 10.3390/plants11030245

**Published:** 2022-01-18

**Authors:** Hafize Fidan, Stanko Stankov, Magdalena Stoyanova, Zhana Petkova, Nadezhda Petkova, Albena Stoyanova, Sezai Ercisli, Ravish Choudhary, Rohini Karunakaran

**Affiliations:** 1Department of Tourism and Culinary Management, Faculty of Economics, University of Food Technologies, 4000 Plovdiv, Bulgaria; docstankov@gmail.com; 2Department of Analytical Chemistry, Technological Faculty, University of Food Technologies, 4000 Plovdiv, Bulgaria; magdalena.stoianova@abv.bg; 3Department of Chemical Technology, University of Plovdiv Paisii Hilendarski, 4000 Plovdiv, Bulgaria; zhanapetkova@uni-plovdiv.net; 4Department of Organic Chemistry and Inorganic Chemistry, Technological Faculty, University of Food Technologies, 4000 Plovdiv, Bulgaria; petkovanadejda@abv.bg; 5Technology of Tobacco, Sugar, Vegetable and Essential oils, Technological Faculty, University of Food Technologies, 4000 Plovdiv, Bulgaria; aastst@abv.bg; 6Department of Horticulture, Atatürk University, Erzurum 25240, Turkey; sercisli@gmail.com; 7Division of Seed Science and Technology, ICAR-Indian Agricultural Research Institute, New Delhi 110012, India; ravianu1110@gmail.com; 8Unit of Biochemistry, Faculty of Medicine, Centre of Excellence for Biomaterials Engineering, AIMST University, Semeling, Bedong 08100, Malaysia

**Keywords:** black pine, *Pinus nigra*, chemical characterization

## Abstract

The present paper aims to investigate the chemical composition of unripe black pine seeds obtained from Bulgaria. The lipid fraction was evaluated in unripe seeds, and the cellulose, total carbohydrates, glucose, fructose, and sucrose were evaluated in seedcakes. The major fatty acid identified in the *Pinus nigra* seed oil was unsaturated linoleic acid (44.2%), followed by the saturated palmitic acid (31.2%). The amount of unsaturated pinolenic (10.5%) and oleic acids (8.8%) was also rather high. The amino acid composition of the protein fraction of seedcakes was also determined. The amino acid composition was represented mainly by asparagine (3.92 mg/g), serine (3.79 mg/g), alanine (3.65 mg/g), arginine (3.32 mg/g), phenylalanine (2.98 mg/g), lysine (2.85 mg/g), proline (2.69 g/mg), tryptophan (2.44 mg/g), valine (2.33 mg/g), isoleucine (2.28 mg/g), and tyrosine (2.05 mg/g). The mineral content (N, P, K, Mg, Na, and Cu) of the seedcakes was evaluated, as the amount of K (8048.00 mg/kg) and Mg (172.99 mg/kg) were the highest in the samples. These findings emphasized the potential use of the unripe black pine seeds in different areas due to their chemical importance and values.

## 1. Introduction

The ability of plants to interact with several biotic and abiotic factors is known as plant ecology. It is the basis of the increasing diversity of plant species, for which the number is over 400,000. It is known that abiotic environmental factors such as humidity, temperature, and sunlight have favorable or adverse effects on plant diversity [1,2]. Plant species have increasing importance for people due to their use in the various spheres, such as pharmaceutical, food, and cosmetic industries, which draw attention to some potential plant species containing biologically active substances. In-depth knowledge of the composition and properties of the plants’ anatomical parts, differences in the design of the biologically active profile relative to the area’s geographical characteristics, methods of analysis, and others, are among the main areas of knowledge of their application [3,4].

The Pinaceae family consists of 11 genera [5]. With more than 100 existing species, *Pinus* is the largest genus of conifers and the most widespread in the Northern Hemisphere. There are five species naturally growing in Bulgaria: *Pinus sylvestris* L., *Pinus nigra* Arn., *Pinus peuce* Grab., *Pinus heldreichii* H., and *Pinus mugo* Turra. Species of the genus *Pinus* are a source of seeds that contain a wide variety of nutrients [6]. Examples of such are cedar seeds, which are obtained from the species *P. pinea* L., and are used as delicacies. The content of monounsaturated and polyunsaturated fatty acids in the composition of seeds of the genus *Pinus* is high, and it is considered a prerequisite for preventing cardiovascular disease [7]. The content of polyunsaturated fatty acids, whose profiles have useful chemometric data for taxonomy and phylogeny of this division, is essential for their application in biomedical and food systems [8]. Many studies on the composition of other species of the genus *Pinus* showed that their composition varies depending on geographical and climatic conditions [8,9,10,11]. The studies showed that pine seeds contain *α*-linolenic acid, antioxidants, and other biologically active components [3,9,10]. The chemical composition of the various anatomical parts of *P. sylvestris*. and *P. nigra,* and the period of their vegetation, determine the composition of many components in their composition. The species’ quantitative and qualitative composition depends on the soil, climatic, geographical, and species characteristics, to no small extent. Both low and high levels of nutrients in plant cells can indicate the species’ resilience and the possibility of its development [12]. 

The use of unripe cones in various food products such as jams, jellies, and infusions in traditional Bulgarian folk medicine necessitates the further evaluation of their chemical characterization. Seeds are a source of biologically active components and may have applications in nutrition or food technology. To the best of our knowledge, there are no data in the literature on the chemical composition of unripe seeds of *P. nigra* (black pine). Therefore, the aim of the present study was to evaluate the chemical profile of unripe seeds of *P. nigra* as a source of biologically active components.

## 2. Results

The moisture content of the unripe seeds of *P. nigra* was 39.44%.

### 2.1. Fatty Acid and Tocopherol Composition of the P. nigra Seed Oil

The lipid fraction of unripe seeds of *P. nigra* was 1.68%. The fatty acid composition of the *P. nigra* unripe seed oil is presented in Table 1.

The amount of essential linoleic (ω-6) fatty acid (44.2%) was the highest in the composition of unripe seed oil. Among the saturated fatty acids, palmitic acid has the highest (31.2%) content. Pinolenic acid is a polyunsaturated fatty acid mainly found in plants, particularly gymnosperms, and it was high in the analyzed seeds at 10.5%. The oleic acid content was lower (8.8%), followed by a small quantity of linolenic acid (3.0%). The content of the rest of the fatty acids ranged from 0.1 to 0.6%. The oil from unripe seeds of *P. nigra* included higher levels of unsaturated fatty acids (67.2%) compared to a lower amount of saturated acids (32.8%).

The amount of polyunsaturated fatty acids predominated (57.8%) in the *P. nigra* unripe seed oil, representing 86.01% of the amount of unsaturated fatty acids (Table 1). The levels of monounsaturated fatty acids were lower (9.4%), and constituted 13.99% of the share of unsaturated fatty acids. 

In terms of the tocopherol composition of the studied *P. nigra* seed oil, *a*-tocopherol (53.1 ± 0.4%) and γ-tocopherol (46.9 ± 0.2%) were detected in the tocopherol fraction. The total tocopherol content of the examined lipids from unripe seeds was significantly high, at 1290 mg/kg in the oil. 

### 2.2. Carbohydrates, Crude Cellulose, Protein, and Amino Acids in the Seedcakes 

The main carbohydrate found in the seedcake was cellulose (28.58 ± 0.27%). 

The soluble sugars in the seedcakes were 0.20 ± 0.02%, represented by glucose (0.09%), fructose (0.02%), and sucrose (0.03%). In our study, sorbitol was not detected in the samples. 

The protein content in the seedcakes was 38.42 ± 0.37%.

The amino acid profile of the protein fraction showed the presence of some essential amino acids (Table 2).

It was found that the share of essential amino acids in the composition of seedcakes of *P. nigra* was represented mainly by asparagine (3.92 mg/g), serine (3.79 mg/g), alanine (3.65 mg/g), arginine (3.32 mg/g), phenylalanine (2.98 mg/g), lysine (2.85 mg/g), tryptophan (2.44 mg/g), valine (2.33 mg/g), isoleucine (2.28 mg/g), and tyrosine (2.05 mg/g). 

### 2.3. Ash and Minerals in Seedcakes

The ash content (2.99 ± 0.02%) of *P. nigra* seedcakes was determined. 

The mineral composition of the *P. nigra* seedcakes is presented in Table 3. The results showed high values of macrominerals in the composition of seedcakes. The relatively diverse composition of micro- and macro-elements is due to diversities in the plant ecology.

Potassium was the mineral with the highest concentration (8048.00 mg/kg), followed by magnesium (172.99 mg/kg). Lower levels of nitrogen (1.96 mg/kg) and phosphorus (0.08 mg/kg) explained the slower rate of plant development, as well as the lower levels of macronutrients in the soil layer. 

## 3. Discussion

The *P. nigra* unripe seeds were characterized with a higher moisture value than the other *Pinus* seed species [6,7]. The higher values of the reported moisture of the unripe seeds could be explained by the embryo’s normal vegetative development of dissolved nutrients before the onset of maturity and the reproductive process.

The amount of the lipid fraction of *P. nigra* unripe seeds in this study was lower than the values obtained from seeds of *P. pinea* (44.9%) [6,8], which showed that the plant ecology, as well as the degree of vegetation and the species type, determined the chemical profile of the seeds.

The results about the fatty acid composition in our study were in agreement with those reported by other researchers [8,13] who analyzed the fatty acid profile of the oils from mature seeds of various *Pinus* species. These previous studies confirmed that linoleic acid was the primary fatty acid in the lipid fraction of the examined species. It is obvious that the oil from *P. nigra* unripe seeds have a similar qualitative, but slightly different quantitative fatty acid composition for the glyceride oil from ripe seeds. The analysis of the oil from unripe *P. nigra* seeds also showed high levels of palmitic acid, which was not reported in the composition of *Pinus* species’ ripe seeds. The amounts of pinolenic and oleic acids in the oil from unripe seeds were lower than the content of the same acids in the lipid fraction of ripe seeds (17.90−18.89% and 16.5−17.8%, respectively) [13,14]. Compared with other *Pinus* species, it was observed that the content of pinolenic acid was similar to that of *P. pinaster* (7.13%) and higher than that from *P. pinea* (0.35%) [13]. According to Wang et al. [15], the content of the above-mentioned fatty acids varied from 0.35% (in *P. pinea*) to 18.77% (in *P. mugol × pimilio*). They also detected small amounts of sciadonic acid in various pine nuts, from 0.87% (*P. koraiensis*) to 4.49% (*P. patula*). Sciadonic acid was not present in the examined *P. nigra* seed oil, which was probably due to the fact that the seeds were unripe. It is established that the content of unsaturated fatty acids increased during the development of the seeds, while those of the saturated ones decreased [16]. The amount of linolenic acid in the present study was two to three times higher than those from the seeds of ripe *Pinus* species—from 0.17 to 1.30% in *P. pinaster*, *P. pinea*, *P. koraiensis*, *P. griffithii*, *P. mughus,* and *P. sylvestris* [13]. Some of the potential benefits to human health due to the ability of pinolenic acid to improve lymphocyte function and its anti-inflammatory action have been previously revealed. These values confirmed the rather balanced fatty acid composition of unripe *P. nigra* seeds and reveals them as a source of essential fatty acids and biologically active components [14]. The total content of the unsaturated fatty acids in the oil from the unripe *P. nigra* seeds was lower than that reported by Bağcı and Karaağaçlı [14], who examined the fatty acid composition in seeds from different Turkish *Pinus* species (from 85.0% in *P. radiata* to 92.1% in *P. sylvestris*). The total saturated fatty acids of the unripe *P. nigra* seeds were mainly due to the high amount of palmitic acid in the fraction. On the other hand, the ω-6/ω-3 ratio was 18.3/1, which was higher than the recommended values (from 2/1 to 5/1) [17]. Generally, the lower the ratio of ω-6/ω-3 fatty acids, the more favorable the oil was at preventing some chronic illnesses. Compared to the fatty acids in the composition of *P. koraiensis* nuts [18], the fatty acid profile of unripe black pine seeds had a lower content of unsaturated fatty acids. In the composition of *P. koraiensis*, 88–89% are unsaturated fatty acids, represented mainly by linoleic (45–46%) and oleic 26–27% acids. The high content of unsaturated fatty acids determines the high degree of permeability through the cell membrane, which places them in the group of essential lipids for the body. The content of saturated fatty acids in the composition of oils has a positive effect on the oxidative processes. The presence of high levels of palmitic and stearic fatty acid also determines the oxidative stability of the lipid fraction. The high levels of polyunsaturated fatty acids in *P. nigra* oil from unripe seeds determines the possibility of their bioactive potential.

The total tocopherol content of the examined unripe sample was significantly higher than most of the seed oils that can be used for human consumption, such as sunflower oil (high oleic) (450–1120 mg/kg), grapeseed oil (240–410 mg/kg), and sesame seed oil (330–1010 mg/kg) [19]. The high levels of the tocopherol fraction in the composition of unripe seeds of *P. nigra* can be used as a natural antioxidant or synergist of other antioxidants used in the food industry. The individual tocopherol composition of the unripe *P. nigra* seed oil was slightly different from that reported by Bağcı and Karaağaçlı [14], where γ-tocopherol predominated in the fraction, followed by α-tocopherol. The content of tocopherols in the composition of unripe seeds of *P. nigra* was lower than the content of tocopherols and tocotrienols reported in cedar nut oil (27.65–37.52 mg/100g) [20]. According to the results obtained in this study, the reported content of tocopherols and tocotrienols in *P. sibrica* nut oil (46.6–49.6 mg/100 g) was higher [21]. The possibility of using secondary raw materials as a potential nutrient source in human and animal nutrition has led to the need for seedcakes samples to be further analyzed. 

The sugar content was lower than that reported previously [6,7] for ripe *P. pinea* seeds (5.15–10.4%). This can be explained both by the species difference and the level of maturity of the seeds. The main monossaccharide was glucose, which corresponds to the findings of Vivas et al. [22] for other *Pinus* species. Moreover, the high cellulose content in seedcakes revealed their potential as a source of dietary fibers.

The protein content in the composition of *P. nigra* seedcakes was higher than the previously reported protein values for the *P. pinea* species, ranging between 31.1–31.6% [6,7]. The high levels of essential amino acids, which form almost half of the amino acids, determine the seedcakes’ high biologically active potential.

The ash content of *P. nigra* seedcakes was lower than that of the *P. pinea* seeds (4.3–4.5%) reported by other authors [6,7].

The relatively diverse composition of micro- and macro-elements is due to differences in climatic conditions and species diversity [12]. Despite the great interest of the scientific community in studying the chemical composition of *Pinus* seed species, no one, as far as we know, has studied the mineral composition of *P. nigra* seeds. The higher levels of potassium and magnesium contained in the *P. nigra* seedcakes confirmed the earlier results obtained in the study of *P. nigra* and *P. sylvestris* seedcakes [12]. Magnesium is a crucial element in many enzymes for the metabolism and the use of calcium. It plays a role in muscle contractions. Magnesium deficiency may cause muscle weakness [23]. On the other hand, K is needed to maintain water−salt balance and alkaline−acid balance in the human body. It plays a role in conducting nerve impulses, muscle contractions, and heart activity [24]. Based on magnesium’s many functions within the human body, it plays an essential role in preventing and treating many diseases, so *P. nigra* seedcakes could be considered a probable source of macro-minerals in several food compositions. The copper content in the *P. nigra* seedcake composition was higher than its content in the *P. nigra* seedcakes (6.6–7.3 mg/kg) obtained in [25], which could be explained by the different vegetative stages, soil conditions, and anatomical factors. According to Nagaiyoti et al. [25], higher levels of micronutrients in the composition of coniferous species may be due to their high soil content due to pollution [25,26].

The inclusion of foods high in potassium and magnesium in the daily diet can improve the cardiovascular system’s activity and lower blood pressure [27]. The creation of new "functional foods" involving many “natural supplements” with a balanced mineral composition can be used as a preventive measure to combat 21st century diseases.

Environmental factors can be divided into climatic, geographical, and soil factors. Previous studies have shown that the chemical composition of plant organs and their active substances are the result of the interaction between plants and the environment in the long process of evolution. The content of some substances can increase significantly under certain environmental conditions such as altitude, average annual temperature, soil characteristics, precipitation, and sunshine [28,29,30,31,32,33,34].

## 4. Materials and Methods

### 4.1. Plant Material 

The samples were collected from the Central Rhodopes (village of Yavrovo, 42° N 24.8° E), Bulgaria. A sampling permit was obtained from the Bulgarian National Science Fund, Project. KP-06-H36/14 from 17.12.2019, managed by Semerdjieva, issued to Tzenka Radoukova and Valtcho D. Zheljazkov. The collected samples were identified as *Pinus nigra* Arn. by Dr. Ivanka Semerdjieva (University of Agriculture, Plovdiv, Bulgaria), according to morphological features such as the color of the plant bark, the color and length of the needles, and the color of the cones [5]. The climate in the region is transitional-continental, brown forest soils predominate, and in some places, there is humus-carbonate. The soil layer is poorly developed and eroded. The terrain is situated on a northern and northeastern slope with alkaline and calcareous rocks. Humidity varies throughout the growing season, from low to relatively moderate. The unripe cones (Figure 1a) were cut with a scalpel, and (manually) the seeds (Figure 1b) were separated from the cotyledons. The seeds (60 g) were placed in plastic containers with lids and stored at 4 ± 2 °C for 10 days before analysis.

Before analysis, the seeds were ground using an electric blender (Bosch MKM 6003, Stuttgart, Germany) to particle sizes of 0.5 mm for 30 s. The sample’s initial moisture content was determined by drying at 103 ± 2 °C to a constant weight, and all of the results were expressed as a dry weight (dw) basis [35].

### 4.2. Determination of Fatty Acids and Tocopherols in the Lipid Fraction

The lipid fraction was extracted from the ground sample (10 g) using n-hexane in a Soxhlet apparatus for 8 h. The solvent was partially removed in a rotary vacuum evaporator. The residue was transferred in a pre-weighed glass vessel. The rest of the solvent was removed under a stream of nitrogen to determine the oil content [36].

#### 4.2.1. Fatty Acid Composition

The fatty acid composition of the triacylglycerols was determined by gas chromatography (GC) [37]. Fatty acid methyl esters (FAMEs) were prepared by pre-esterification of the triacylglycerols with sulfuric acid in methanol [38]. The determination of FAMEs was performed on a HP 5890 gas chromatograph (Mundelein, IL, USA) equipped with a 75 m × 0.18 mm × 25 μm (film thickness) capillary Supelco column and a flame ionization detector. The column temperature was from 140 °C (held for 5 min), at 4 °C/min to 240 °C (held for 3 min); the injector and detector temperatures were set at 250 °C. Identification was performed by comparison of the retention times with those of a standard mixture of FAME (Supelco 37 comp. FAME mix – Sigma Aldrich, Darmstadt, Germany) and a standard of pinolenic acid methyl ester (Toronto Research Chemicals, Toronto, ON, Canada) subjected to GC under identical experimental conditions. 

#### 4.2.2. Tocopherols

The tocopherols were determined by the previously described method [39], on a Merck-Hitachi (Merck, Darmstadt, Germany) high performance liquid chromatograph instrument equipped with 250 mm × 4 mm Nucleosil Si 50-5 column and fluorescent detector Merck-Hitachi F 1000. The operating conditions were a mobile phase of n-hexane/ dioxane of 96:4 (*v*/*v*) and a flow rate of 1 mL/min, excitation of 295 nm, and emission of 330 nm. Then, 20 μL 2% solution of crude oil in n-hexane was injected. The tocopherols were identified by comparing the retention times with those of the authentic individual ones. The tocopherol content was calculated on the base of tocopherol peak areas in the sample vs. tocopherol peak area of a standard tocopherol solution (DL-α-, DL-β-, DL-γ-, and DL-δ-tocopherols, purchased from Merck KGaA, Darmstadt, Germany). The limit of detection in HPLC was 0.0016 µg/mL.

### 4.3. Determination of Carbohydrates, Protein, and Amino Acids in the Seedcakes

The total soluble carbohydrate content was evaluated by the phenolsulfuric acid method using 5 g of the ground seed sample for the analysis [40].

HPLC-RID analysis of the sugars was performed according to the method described by Petkova et al. [41].

Crude cellulose was determined using the method described by [42].

The total protein content was analyzed according to the method of [43] with a UDK 152 Kjeldahl System (Velp Scientiffica, Via Strasione, Italy).

The amino acid composition was determined by the method previously described by Stankov et al. [44].

### 4.4. Determination of Ash and Mineral Content in the Seedcakes

The ash content of the sample was determined by igniting the sample at 600 °C for 5 h.

The air-dried sample was mineralized at 450 °C. The residue was first dissolved in concentrated HCl and evaporated to dryness. Then, the residue was dissolved in 0.1 mol/L HNO_3_ solution. Mineral contents were determined on an atomic absorption spectrophotometer (AAS) Perkin Elmer/HGA 500 (Norwalk, USA), under the following instrumental parameters for the flame AAS: phosphorus (P) 470 nm; sodium (Na) 589.6 nm; potassium (K), 766.5 nm; magnesium (Mg), 285.2 nm; calcium (Ca), 317.0 nm; and copper (Cu), 324.7 nm. The identification of metals was carried out by comparison with a standard solution of metal salts, and the metal concentrations were calculated from a calibration curve, built using a standard 1 μg/mL solution [45]. The nitrogen (N) content was determined by the method described in [46].

### 4.5. Statistical Analysis 

The measurements were performed in triplicate, and the results were presented as the mean value of the individual measurements with the corresponding standard deviation (SD), using Microsoft Excel 2010.

## 5. Conclusions

In the present study, the chemical composition of *P. nigra* Arn. unripe seeds obtained from Bulgaria was evaluated. The high proportion of tocopherols (1290 mg/kg), the high ratio (44.97%) of essential amino acids, and the high levels of the micronutrients potassium and magnesium make the studied plant as a secondary material a possible alternative source or component in the composition of functional foods. The results obtained in this study revealed that the plant ecology highly determined the composition of the seeds. The authors are aware that our research may have some limitations. The effect of the seasonal, genotypic, and environmental variability of the chemical contents in *P. nigra* needs to be observed. Therefore, this preliminary study marks the beginning of new, more in-depth analyses of *P. nigra* ’s potential uses in food technology. Further studies are required in order to determine the cost, applicability, and safety of the studied samples. Our future studies will be focused on the composition of the products obtained with the participation of unripe cones. We believe that it will complement the knowledge of the composition of both raw materials and finished products.

## Figures and Tables

**Figure 1 plants-11-00245-f001:**
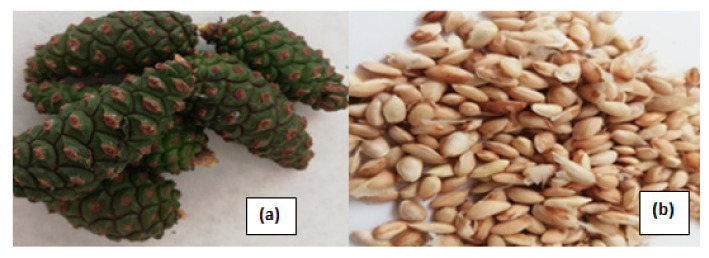
Unripe cones (**a**) and seeds (**b**) of *P. nigra*.

**Table 1 plants-11-00245-t001:** Fatty acid and tocopherol composition of *P. nigra* unripe seed oil (mean ± SD).

№	Fatty Acids, %		Content
1.	Capric acid	C _10:0_	0.1 ± 0.0
2.	Undecylic acid	C _11:0_	0.1 ±0.0
3.	Myristic acid	C _14:0_	0.6 ± 0.0
4.	Myristoleic acid	C _14:1_ _(9)_	0.2 ± 0.0
5.	Pentadecilyc acid	C _15:0_	0.6 ± 0.1
6.	Palmitic acid	C _16:0_	31.2 ± 0.2
7.	Palmitoleic acid	C _16:1_ _(9)_	0.4 ± 0.1
8.	Margaric acid	C _17:0_	0.1 ± 0.0
9.	Stearic acid	C _18:0_	0.1 ± 0.0
10.	Oleic acid	C _18:1_ _(9)_	8.8 ± 0.2
11.	Linoleic acid	C _18:2_ _(9,12)_	44.2 ± 0.4
12.	Pinolenic acid	C _18:3_ (_5,9,12_)	10.5 ± 0.4
13.	Linolenic acid	C _18:3_ (_9,12,15_)	3.0 ± 0.2
14.	Eicosadienoic acid	C _20:2_	0.1 ± 0.0
Saturated fatty acids			32.80 ± 0.01
Monounsaturated fatty acids			9.40 ± 0.02
Polyunsaturated fatty acids			57.80 ± 0.01
**Tocopherols**			**Content**
*α*-Tocopherol, % (of total tocopherols)			53.1 ± 0.4
γ-Tocopherol, % (of total tocopherols)			46.9 ± 0.2
Total tocopherols, mg/kg			1290 ± 17

**Table 2 plants-11-00245-t002:** Amino acid composition of the protein fraction of *P*. *nigra* seedcakes (mean ± SD), mg/g.

Amino Acid	Content, mg/g
Asparagine	3.92 ± 0.03
Serine	3.79 ± 0.03
Glutamic acid	3.00 ± 0.02
Glycine	1.73 ± 0.01
Histidine	0.05 ± 0.0
Arginine	3.32 ± 0.03
Thryptophan	2.44 ± 0.01
Alanine	3.65 ± 0.03
Proline	2.69 ± 0.01
Cysteine	0.02 ± 0.0
Tyrosine	2.05 ± 0.01
Valine	2.33 ± 0.01
Methionine	0.43 ± 0.0
Lysine	2.85 ± 0.01
Isoleucine	2.28 ± 0.01
Leucine	0.45 ± 0.01
Phenylalanine	2.98 ± 0.01

**Table 3 plants-11-00245-t003:** Macro- and microelements concentrations in the seedcakes of *P. nigra*.

Macroelements	Content, mg/kg	Microelements	Content, mg/kg
Nitrogen (N)	1.96	Copper (Cu)	21.02
Phosphorus (P)	0.08		
Potassium (K)	8048.00		
Magnesium (Mg)	172.99		
Sodium (Na)	6.76		
Calcium (Ca)	*		

* not detected.

## Data Availability

All new research data were presented in this contribution.

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
