# Peer review of "Chemical Composition of Pinus nigra Arn. Unripe Seeds from Bulgaria"

_plants, 2022, doi:10.3390/plants11030245_

Round 1
Reviewer 1 Report
The manuscript entitled "Phytochemical Composition of Pinus nigra Arn. Unripe Seeds from Bulgaria" reports a phytochemical study of unripe black pine seeds obtained from Bulgaria. There are some points to increase the visibility of the manuscript.
1 - Please, it is suitable to add a figure with the structures of the fatty acids cited in table 1.
2 - Please, improve the quality of figure 2.
3 - Abstract
"These findings 32 emphasized the potential use of the unripe black pine seeds as an alternative source of bioactive components in different areas due to their phytochemical importance and values."
Please, there isn't any information regarding bioactive components reported. Please clarify, rewriting the abstract.
4 - Discussion and Conclusion
"Seeds as a source of biologically 289 active components may find application in nutrition or food technology"
Please, it is suitable to improve the discussion of the bioactive properties of some components reported in the study that corroborates this statement.
Author Response
Reviewer#1
The manuscript entitled "Phytochemical Composition of Pinus nigra Arn. Unripe Seeds from Bulgaria" reports a phytochemical study of unripe black pine seeds obtained from Bulgaria. There are some points to increase the visibility of the manuscript.
Q1) Please, it is suitable to add a figure with the structures of the fatty acids cited in table 1.
A1) Table 1 was enlarged
Q2) Please, improve the quality of figure 2.
A2) The figure 2 was removed due to the requirement of the reviewer
Q3) Abstract
"These findings 32 emphasized the potential use of the unripe black pine seeds as an alternative source of bioactive components in different areas due to their phytochemical importance and values."
Please, there isn't any information regarding bioactive components reported. Please clarify, rewriting the abstract.
A3) The sentence was revised.
Q4) Discussion and Conclusion
"Seeds as a source of biologically 289 active components may find application in nutrition or food technology"
Please, it is suitable to improve the discussion of the bioactive properties of some components reported in the study that corroborates this statement.
A4) Discussion part was improved and the specific sentence was deleted.
Reviewer 2 Report
- Why did the authors decide to study unripe pine nuts. Is there any economic value or can unripe seeds be used as a food?
- Resolution of Figure 2 can be improved.
- The FAME standard mix does not contain pinolenic acid. How was it identified? Some mass spectrometric methods can clearly identify unusual FAME such as pinolenic acid and sciadonic acid (20:3(5Z,11Z,14Z)). A statement or justification must be given if unusual FAME were identified tentatively. Also, sciadonic acid is present in significant amounts in various Pinus spp (See reference: Anal. Chem. 2020, 92, 8209−8217). Please indicate in the article if efforts were given to identify it.
- In discussion, the authors can compare the fatty acid profile, etc. of P nigra to other common Pinus spp. used as food.
- English doesn’t flow. Please make extensive efforts to improve grammar and polish sentences.
Author Response
Reviewer#2
Q1) Why did the authors decide to study unripe pine nuts. Is there any economic value or can unripe seeds be used as a food?
A1) There are no data in the literature on humans' independent consumption of unripe pine nuts. In Bulgaria, immature cones (along with the cones) in jams, jellies, and syrups is an old practice. Commonly, folk medicine recommends consuming syrups from green cones for colds and disorders of the respiratory system.
Q2) Resolution of Figure 2 can be improved
A2) The figure was removed due to the reviewer’s requirement
Q3) The FAME standard mix does not contain pinolenic acid. How was it identified? Some mass spectrometric methods can clearly identify unusual FAME such as pinolenic acid and sciadonic acid (20:3(5Z,11Z,14Z)). A statement or justification must be given if unusual FAME were identified tentatively. Also, sciadonic acid is present in significant amounts in various Pinus spp (See reference: Anal. Chem. 2020, 92, 8209−8217). Please indicate in the article if efforts were given to identify it.
A3) For determination of the retention time of the pinolenic acid was used a standard of pinolenic acid methyl ester (from TRC Canada). This was added in the article:
“Identification was performed by comparison of the retention times with those of a standard mixture of FAME (Supelco, USA 37 comp. FAME mix) and a standard of pinolenic acid methyl ester (TRC Canada) subjected to GC under identical experimental conditions.”
Sciadonic acid was not detected in the sample.
Q4) In discussion, the authors can compare the fatty acid profile, etc. of P nigra to other common Pinus spp. used as food.
A4) The fatty acid profile of the oil from P. nigra’ seeds was compared to other Pinus spp.
‘The analysis of the oil from unripe P. nigra seeds also showed high levels of palmitic acid (31.2%), which was not reported in the composition of Pinus species' ripe seeds. The amounts of pinolenic (10.5%) and oleic (8.8%) acids in the oil from unripe seeds were lower than the content of the same acids in the lipid fraction of ripe seeds (17.90 - 18.89% and 16.5 - 17.8%, respectively) [13, 14]. Compared with other Pinus species could be observed that the content of pinolenic acid was similar to that of P. pinaster (7.13%) and higher than those from P. pinea (0.35%) [13]. The amount of linolenic acid in the present study was two to three times higher than those from the seeds of ripe Pinus species – from 0.17 to 1.30% in P. pinaster, P. pinea, P. koraiensis, P. griffithii, P. mughus and P. sylvestris [13]. Some of the potential benefits to human health due to the ability of pinolenic acid to improve lymphocyte function and its anti-inflammatory action were previously revealed. These values confirmed the balanced fatty acid composition of unripe P. nigra seeds as a source of essential fatty acids and biologically active components [14]. Total content of the unsaturated fatty acids in the oil from the unripe P. nigra seeds was lower than reported by BaÄŸcı and KaraaÄŸaçlı [14] who examined the fatty acid composition in seeds from different Turkish Pines (from 85.0% in P. radiata to 92.1% in P. sylvestris). Total saturated fatty acids of the unripe P. nigra seeds was mainly due to the high amount of palmitic acid in the fraction. On the other hand, the n-6/n-3 ratio was 18.3/1 which was higher than the recommended values (2/1 to 5/1) [Simopoulos (2002)]. Generally, the lower the ratio of n-6/n-3 fatty acids, the more favorable was the oil in preventing some chronic illnesses.
The high levels of the tocopherol fraction in the composition of unripe seeds of P. nigra can be used as a natural antioxidant or synergist of other antioxidants used in the food industry. The individual tocopherol composition of the unripe P. nigra seed oil was slightly different from the reported by BaÄŸcı and KaraaÄŸaçlı [14] where γ-tocopherol predominated in the fraction, then followed by α-tocopherol.’
Q5) English doesn’t flow. Please make extensive efforts to improve grammar and polish sentences.
A5) The language was improved.
Reviewer 3 Report
In this manuscript the authors present the phytochemical characterization of P. nigra Arn. seeds, as a source of potential use in food technology. The manuscript is interesting, but it needs to be improved.
1) Despite the authors themselves say in the Conclusion section that this study is "preliminary" and "may have some limitations", they should explain why they investigated for the first time the unripe seeds, without a comparative characterization of the ripe ones.
2) L 158-159: "balanced fatty acid composition" and "a source of essential fatty acids" are in contrast to the observed high level of palmitic acid, that is a saturated fatty acid and is not an essential fatty acid. Surprisingly, the authors fail to comment on the favorable ratio omega 6/omega 3 fatty acids (ratio less than 4)
3) Table 2 is present twice, the one in L 81 is out of place and therefore needs to be eliminated.
4) Ref. 15 and 16 are identical. Provide the appropriate ref. 16
5) L 165: correct the "monosaccharide" term
6) L 176: change "composition" in "mineral composition"
7) Write "Pinus nigra" in italics along the manuscript (L 79, L 286....)
8) L 85: would "n-6" be "w-6"?
9) L 98-99, 104-106 and 122-123 would be better placed in the Discussion section
10) L 186-187: it is not clear the term of comparison for the copper content
11) Paragraph 4.2.2.: please, specify the tocopherol standards and report the limit of detection in µg/kg
Author Response
Reviewer#3
In this manuscript the authors present the phytochemical characterization of P. nigra Arn. seeds, as a source of potential use in food technology. The manuscript is interesting, but it needs to be improved.
Q1) Despite the authors themselves say in the Conclusion section that this study is "preliminary" and "may have some limitations", they should explain why they investigated for the first time the unripe seeds, without a comparative characterization of the ripe ones.
A1) Unripe cones are used in jams, jellies, and infusions in traditional Bulgarian folk medicine. Our further studies will be focused on the composition of the products obtained with the participation of unripe cones. We believe that it will complement the knowledge of the composition of both raw materials and finished products.
Q2) L 158-159: "balanced fatty acid composition" and "a source of essential fatty acids" are in contrast to the observed high level of palmitic acid, that is a saturated fatty acid and is not an essential fatty acid. Surprisingly, the authors fail to comment on the favorable ratio omega 6/omega 3 fatty acids (ratio less than 4)
A2) The place of the following sentence (‘The analysis of the oil from unripe P. nigra seeds also showed high levels of palmitic acid (31.2%), which was not reported in the composition of Pinus species' ripe seeds.’) was replaced in the paragraph, because the meaning of the next sentence was ambiguous.
Even though the content of palmitic acid is 31.2%, the amount of total polyunsaturated fatty acids is 57.8% which is still high. Also, a brief discussion was added about the ratio omega 6/omega 3 fatty acids.
The paragraph was changed to:
‘The results about the fatty acid composition in the study were in agreement with these reported by other researchers [8, 13], who analyzed the fatty acid profile of the oils from mature seeds of various Pinus species. These previous studies confirmed that linoleic acid was the primary fatty acid in the lipid fraction of the examined species. It is obvious that the oil from P. nigra unripe seeds have a similar qualitative but slightly different quantitative fatty acid composition from the glyceride oil from ripe seeds. The analysis of the oil from unripe P. nigra seeds also showed high levels of palmitic acid (31.2%), which was not reported in the composition of Pinus species' ripe seeds. The amounts of pinolenic (10.5%) and oleic (8.8%) acids in the oil from unripe seeds were lower than the content of the same acids in the lipid fraction of ripe seeds (17.90 - 18.89% and 16.5 - 17.8%, respectively) [13, 14]. Compared with other Pinus species could be observed that the content of pinolenic acid was similar to that of P. pinaster (7.13%) and higher than those from P. pinea (0.35%) [13]. The amount of linolenic acid in the present study was two to three times higher than those from the seeds of ripe Pinus species – from 0.17 to 1.30% in P. pinaster, P. pinea, P. koraiensis, P. griffithii, P. mughus and P. sylvestris [13]. Some of the potential benefits to human health due to the ability of pinolenic acid to improve lymphocyte function and its anti-inflammatory action were previously revealed. These values confirmed the rather balanced fatty acid composition of unripe P. nigra seeds and reveals them as a source of essential fatty acids and biologically active components [14]. Total content of the unsaturated fatty acids in the oil from the unripe P. nigra seeds was lower than reported by BaÄŸcı and KaraaÄŸaçlı [14] who examined the fatty acid composition in seeds from different Turkish Pines (from 85.0% in P. radiata to 92.1% in P. sylvestris). Total saturated fatty acids of the unripe P. nigra seeds was mainly due to the high amount of palmitic acid in the fraction. On the other hand, the n-6/n-3 ratio was 18.3/1 which was higher than the recommended values (2/1 to 5/1) [Simopoulos (2002)]. Generally, the lower the ratio of n-6/n-3 fatty acids, the more favorable was the oil in preventing some chronic illnesses.’
Q3) Table 2 is present twice, the one in L 81 is out of place and therefore needs to be eliminated.
A3) The table was removed
Q4) Ref. 15 and 16 are identical. Provide the appropriate ref. 16
A4) The reference 16 was removed
Q5) L 165: correct the "monosaccharide" term
A5) The revision was made
Q6) L 176: change "composition" in "mineral composition"
A6) The revision was made
Q7) Write "Pinus nigra" in italics along the manuscript (L 79, L 286....)
A7) The revision was made
Q8) L 85: would "n-6" be "w-6"?
A8) The required correction was made.
Q9) L 98-99, 104-106 and 122-123 would be better placed in the Discussion section
A9) The revision was made
Q10) L 186-187: it is not clear the term of comparison for the copper content
A10) The revision was made
Q11) Paragraph 4.2.2.: please, specify the tocopherol standards and report the limit of detection in µg/kg
A11) The standard tocopherol solution (DL-α-, DL-β-, DL-γ-, and DL-δ-tocopherols) was purchased from Merck KGaA, Darmstadt, Germany. The limit of detection in HPLC was 0.0016 µg/mL. These changes were incorporated in the Paragraph 4.2.2.
Reviewer 4 Report
In the present study, the authors evaluated some phytochemical composition of Pinus nigra Arn. unripe seeds obtained from Bulgaria.
The work has several oversights and needs to be better organized.
The main revisions are reported below
- Table 2 is shown twice in the text, there is no tables for carbohydrates, soluble sugars and tocopherols
- In table 1 delete the dash in C18: 3 (5,9,12-)
- delete graph 2 and report percentages of SFA, MUFA and PUFA at the bottom of table 1
- As the result values are often reported in the discussion, I would suggest to combine the results and discussion in one section
- Lines from 160 to 162, on which basis of data reported in the literature did you establish that the tocopherol values ​​are high
- In materials and methods section report the growth conditions of Pisus nigra
- In paragraph 4.2 report the amount of the sample used for the extraction and the volume of solvent added
- Line 269: uniform mL-1 with others unit of measurement reported in the text
- In the conclusions section, i think that it is excessive to report that “unripe seeds of P. nigra contained a rich amount of biologically active components” since only few classes of bioactive molecules have been analyzed.
Author Response
Reviewer#4
In the present study, the authors evaluated some phytochemical composition of Pinus nigra Arn. unripe seeds obtained from Bulgaria.
The work has several oversights and needs to be better organized.
The main revisions are reported below
Q1) Table 2 is shown twice in the text, there is no tables for carbohydrates, soluble sugars and tocopherols
A1) The revision was made
Q2) In table 1 delete the dash in C18: 3 (5,9,12-)
A2) The dash was deleted.
Q3) Delete graph 2 and report percentages of SFA, MUFA and PUFA at the bottom of table 1
A3) The required corrections were made
Q4) As the result values are often reported in the discussion, I would suggest to combine the results and discussion in one section
A4) Thank you for your suggestion but the section Results and Discussion was given separately due to the journal’s requirements.
Q5) Lines from 160 to 162, on which basis of data reported in the literature did you establish that the tocopherol values ​​are high
A5) The comparison was with some seed oils which could be used for human consumption, such as sunflower oil (high oleic), grapeseed oil and sesameseed oil.
‘Total tocopherol content of the examined unripe sample was significantly higher than most of the seed oils which could be used for human consumption, such as sunflower oil (high oleic) (450-1120 mg/kg), grapeseed oil (240-410 mg/kg) and sesameseed oil (330-1010 mg/kg) [Codex-Stan 210 – 1999].’
Q6) In materials and methods section report the growth conditions of Pisus nigra
A6) The climate in the region is transitional-continental, brown forest soils predominate, and in some places, there are humus-carbonate. The soil layer is poorly developed and eroded. The terrain is situated on a northern and northeastern slope with alkaline and calcareous rocks. Humidity varies throughout the growing season, from low to relatively moderate.
Q7) In paragraph 4.2 report the amount of the sample used for the extraction and the volume of solvent added
A7) The revision was made
Q8) Line 269: uniform mL-1 with others unit of measurement reported in the text
A8) The revision was made
Q9) In the conclusions section, i think that it is excessive to report that “unripe seeds of P. nigra contained a rich amount of biologically active components” since only few classes of bioactive molecules have been analyzed.
A9) The sentence was removed
Round 2
Reviewer 2 Report
There are some improvements of the manuscript but I don’t think the current quality meets the requirements for publication.
1. There are still many weird sentences throughout the text.
Just use the following as an example:
“Pinolenic acid is a plant-based polyunsaturated fatty acid with a high amount in the unripe seeds' composition (10.5%).”
What is “plant-based”? We are not talking about a food such as vegan burger.
Consider modifying it to something like “Pinolenic acid is a polyunsaturated fatty acid mainly found in plants particularly gymnosperms, and it is high in the analyzed seeds at 10.5%.”
2. Please provide a few chromatograms (in supplements) since the authors claimed that they used pinolenic acid standards and also did not find sciadonic acid. The absence of sciadonic acid may be a result of low sensitivity or concentration of the samples. Sciadonic acid was found in many Pinus spp. (Anal. Chem. 2020, 92, 8209−8217) and at least one early publication also reported its presence in P. nigra at 0.32%. If you really did not detect any sciadonic acid, please refer to these references for a discussion.
Author Response
Q1) There are still many weird sentences throughout the text.
Just use the following as an example:
“Pinolenic acid is a plant-based polyunsaturated fatty acid with a high amount in the unripe seeds' composition (10.5%).”
What is “plant-based”? We are not talking about a food such as vegan burger.
Consider modifying it to something like “Pinolenic acid is a polyunsaturated fatty acid mainly found in plants particularly gymnosperms, and it is high in the analyzed seeds at 10.5%.”
A1) We would like to thank to Review#1 for remarks and recommendations. We seriously considered these remarks and recommendations and we corrected them all on the revised version.
Q2) Please provide a few chromatograms (in supplements) since the authors claimed that they used pinolenic acid standards and did not find sciadonic acid. The absence of sciadonic acid may be a result of the low sensitivity or concentration of the samples. Sciadonic acid was found in many Pinus spp. (Anal. Chem. 2020, 92, 8209−8217) and at least one early publication also reported its presence in P. nigra at 0.32%. If you really did not detect any sciadonic acid, please refer to these references for a discussion.
A2) We would like to thank Reviewer#1 for remarks and recommendations. We seriously considered these remarks and recommendations and we corrected them all on the revised version.
The mentioned literature was used in the Discussion section and the following paragraph was added in the manuscript:
“According to Wang et al. [15] the content of the above-mentioned fatty acids varied from 0.35% (in Pinus pinea) to 18.77% (in Pinus Mugol x pimilio). They also detected small amounts of sciadonic acid in various pine nuts from 0.87% (Pinus koraiensis) to 4.49% (Pinus patula). Sciadonic acid was not present in the examined P. nigra seed oil which was probably due to the fact that the seeds were unripe. It is established that the content of unsaturated fatty acids increases during the development of the seeds, while those of saturated ones decreases [16].”
Reviewer 3 Report
Minor points:
1) Keywords: I suggest to replace "Chemical activity" with "Chemical characterization" or "Chemical profile"
2) L 52 write "Pinus" in Italics
3) I suggest the authors to mention the uses of unripe cones, as described in A1 and added in the Conclusions, in the last part of the Introduction.
The perspectives of the authors added in the Conclusions are not in harmony with the generic potential food applications of the characterized seeds. The authors have to move them elsewhere, or rather harmonize them.
Author Response
Q1) Keywords: I suggest to replace "Chemical activity" with "Chemical characterization" or "Chemical profile"
A1) We would like to thank Review#2 for remarks and recommendations. We corrected it
Q2) L 52 write "Pinus" in Italics
A2) Corrected
Q3) I suggest the authors to mention the uses of unripe cones, as described in A1 and added in the Conclusions, in the last part of the Introduction.
A3) Corrected
Q4) The perspectives of the authors added in the Conclusions are not in harmony with the generic potential food applications of the characterized seeds. The authors must move them elsewhere or harmonize them.
A3) Corrected
Reviewer 4 Report
-Report Saturated fatty acids, Monounsaturated fatty acids and Polyunsaturated fatty acids in the table of the fatty acids composition (table 1)
-There are not tables about tocopherols, carbohydrates and soluble sugars.
-I suggest not to repeat the results in the discussion section
Author Response
Q1) Report Saturated fatty acids, Monounsaturated fatty acids and Polyunsaturated fatty acids in the table of the fatty acids composition (Table 1)
A1) The results about the saturated, mono- and polyunsaturated fatty acids were added in Table 1.
Q2) There are no tables about tocopherols, carbohydrates and soluble sugars.
A2) Corrected. The results about the tocopherols were included in Table 1.
Q3) I suggest not to repeat the results in the discussion section
A3) Corrected